# A Multi-Institutional Database Review of Orbital Complications and Survival Outcomes in Adult Patients with Invasive or Non-Invasive Fungal Rhinosinusitis

**DOI:** 10.3390/jof8121239

**Published:** 2022-11-23

**Authors:** Po-Teng Chiang, Sheng-Dean Luo, Ren-Wen Ho, Ching-Nung Wu, Kuan-Chung Fang, Wei-Chih Chen

**Affiliations:** 1Department of Otolaryngology, Kaohsiung Chang Gung Memorial Hospital and Chang Gung University College of Medicine, Kaohsiung 83301, Taiwan; 2Department of Ophthalmology, Kaohsiung Chang Gung Memorial Hospital and Chang Gung University College of Medicine, Kaohsiung 83301, Taiwan

**Keywords:** invasive fungal rhinosinusitis, orbital complications, fungal rhinosinusitis, non-invasive fungal rhinosinusitis, aspergillus, mucor

## Abstract

Background: Invasive fungal rhinosinusitis (IFS) with orbital complications has remained a challenging disease over the past few decades. Only a few studies have been conducted to investigate the factors associated with orbital complications in fungal rhinosinusitis (FRS). We aimed to review the characteristics between IFS and non-invasive fungal rhinosinusitis (NIFS) and determine clinical factors associated with orbital complications and overall survival. Methods: A multi-institutional database review study was conducted using the Chang Gung Research Database (CGRD) from January 2001 to January 2019. We identified FRS patients using International Classification of Diseases diagnosis codes and SNOMED CT. We categorized patients into IFS and NIFS groups and analyzed the demographic data, underlying diseases, clinical symptoms, laboratory data, image findings, fungal infection status, and survival outcomes. Results: We included 1624 patients in our study, with 59 IFS patients and 1565 NIFS patients. The history of an organ or hematopoietic cell transplantation had a significant prognostic effect on the survival outcomes, with surgical intervention and high hemoglobin (Hb) and albumin levels recognized as positive predictors. Posterior ethmoid sinus involvement, sphenoid sinus involvement, facial pain, blurred vision, and periorbital swelling were risk factors of orbital complications. Conclusions: In NIFS patients, orbital complications were found to be associated with old age, a high WBC count, high blood glucose, and a high CRP level. For the risk factors of orbital complications in IFS patients, posterior ethmoid sinus involvement, sphenoid sinus involvement, facial pain, blurred vision, and periorbital swelling were recognized as predictors. Among IFS patients, a history of organ or hematopoietic cell transplantation was a risk factor for poor survival, while, conversely, surgical intervention and high Hb and albumin levels were related to improved survival. As predictors of orbital complications in IFS patients, posterior ethmoid sinus involvement, sphenoid sinus involvement, facial pain, blurred vision, and periorbital swelling upon the first visit should raise attention, with close monitoring.

## 1. Introduction

Fungal diseases of the paranasal sinus have been reported with an elevating prevalence in recent decades [1,2,3,4]. However, the pathophysiology between fungal colonization of sinonasal mucosa and the development to further fungal infection has not been fully developed and determined. Factors including prolonged antibiotic use, host immune status, underlying disease, and moist environments may contribute to the development of fungal rhinosinusitis [5]. Fungal rhinosinusitis (FRS) is currently recognized as a disease spectrum with distinct pathologic features. Accordingly, several classifications of FRS have been proposed since the twentieth century, with evolvement into the categorization of invasive fungal rhinosinusitis (IFS) or non-invasive fungal rhinosinusitis (NIFS), presented by Hora in 1965 [6]. At present, the generally accepted system approved by international consensus classifies FRS as invasive fungal sinusitis or non-invasive fungal sinusitis by the penetration of mucosa by hyphae [7]. Compared to NIFS, IFS may result in an intense invasion of fungal hyphae through the mucosa, neurovascular structures, bone, and surrounding organs [8,9]. Since the clinical outcomes substantially vary between NIFS and NIFS, establishing a precise diagnosis and using the proper treatment modalities are crucial for FRS patients to achieve a better prognosis and survival [10].

Among the complications within the disease process of FRS, orbital complications are some of the most commonly encountered, with their prevalence ranging from 23.7% to 49.6% in IFS patients [11,12]. Conversely, orbital complications in NIFS have mostly been reported in case reports or case series. Instead, radiological findings of orbital involvement are more commonly discussed in NIFS, with the prevalence ranging from 12.1% to 30% in previous limited studies [13,14,15]. IFS with orbital complications has remained a challenging disease over the past few decades, which often carries a rapidly progressive feature and results in delayed diagnosis. Despite the advancement of endoscopic video systems, histopathologic staining, and diagnostic radiography, it remains a challenge for clinicians to promptly establish a distinct categorization due to the prolonged detection period of fungal cultures and pathology reports, varying experiences in clinical practices, inconsistent culture findings, and possible incompletion of the diagnostic processes owing to the patient’s general condition. Another critical issue for the early differentiation of FRS is the high mortality among patients with IFS, reported to range from 18% to 64.9% across studies [12,16,17,18,19]. Despite the early prescription of antifungal medication and the evolution of surgical techniques, the prognosis remains unsatisfactory for IFS patients, especially for those with poorly controlled diabetes mellitus, autoimmune diseases, hematologic malignancies, or immunosuppression status [12,18,20,21,22,23]. The delay in diagnosis caused by awaiting pathology reports and the findings of fungal cultures often leads to poor prognostic consequences.

Owing to the rarity of IFS, studies that entail a large sample size are relatively challenging. Only a few studies have been conducted to investigate the factors associated with the IFS with orbital complications. Our study aimed to determine clinical factors associated with orbital complications in patients with IFS and NIFS. In addition, the predictors of orbital complications and survival outcomes in IFS were also investigated.

## 2. Materials and Methods

### 2.1. Data Source

We retrieved data from the Chang Gung Research Database (CGRD), which is a de-identified database of the Chang Gung Memorial Hospital (CGMH) in Taiwan. As a medical institution containing seven branches, the CGMH includes several tertiary hospitals distributed widely across the country and comprises patients with a high complexity and severity of diseases. Through coding using the International Classification of Diseases, Ninth Revision (ICD-9) and International Classification of Diseases, Tenth Revision (ICD-10), the database has been categorized by licensed nosologists and used widely with confirmed accuracy [24,25]. The present study was approved by the Institutional Review Board of the Kaohsiung Branch of the Chang Gung Memorial Hospital.

### 2.2. Study Population and Study Design

Data from 1624 patients diagnosed with fungal sinusitis were obtained from the Chang Gung Research Database (CGRD) from January 2001 to January 2019, and a multi-institutional retrospective study was conducted. Patients with an ICD-9 or ICD-10 diagnosis of either sinusitis or nasal polyps were initially identified and narrowed down to the diagnosis of fungal infection using criteria including diagnostic codes of mycosis (ICD-9: 1179 and ICD-10: B49) or aspergillosis (ICD-9: 1173 and ICD-10: B44), positive culture results of fungi, or the presence of fungi in the pathology report (SNOMED CT). We rigorously reviewed the medical records and determined the prescription of antifungal medication during hospitalization as the essential criterion for IFS. Among these limited patients, we further defined patients with IFS as those with invasive features approved by histopathologic findings or fungal sinusitis with a high clinical suspicion of IFS, including ophthalmologic complications or neurological deficits. Different types of fungal infection were confirmed by pathologists according to the histopathology report and immunochemistry staining, classified as mucormycosis, aspergillosis, or other fungal types. All patients who did not undergo surgery or biopsy had culture reports of fungal species, including *Mucor*, *Aspergillus*, or *candida*. These patients were also categorized into the subgroups of mucormycosis, aspergillosis, and other fungal types, respectively. Orbital complications of rhinosinusitis were confirmed by radiologists according to the image findings of computed tomography (CT) or magnetic resonance imaging (MRI) and staged according to Chandler’s classification [26]. Neurologic deficits included blurred vision, ptosis, diplopia, extraocular muscle (EOM) limitation, and altered consciousness. We excluded patients under 18 years of age and those who were not admitted due to fungal sinusitis. The patients were divided into four groups according to the final enrollment of IFS and orbital complications, including IFS with orbital complications (IFSwOC), IFS without orbital complications (IFSsOC), NIFS with orbital complications (NIFSwOC), and NIFS without orbital complications (NIFSsOC). Demographic data, underlying disease, clinical symptoms, laboratory data, image findings, culture reports, fungal infection status, antifungal medications, surgical outcomes, and survival outcomes were collected for further analysis. The risk factors of orbital complications and survival outcomes of IFS were investigated.

### 2.3. Statistical Analysis

All statistical analyses were conducted using SAS version 9.4 (SAS Institute, Cary, NC, USA). For the differences between subgroups, the comparisons between categorical variables were analyzed using chi-square tests or Fisher exact tests. The differences between continuous variables were derived from independent Student’s *t*-tests or Mann–Whitney *U* tests. The relationship between the study variables and survival was evaluated using univariate Cox regression models. Orbital complications were analyzed through univariable logistic regression. A *p*-value < 0.05 was considered statistically significant.

## 3. Results

### 3.1. Patient Characteristics

Between January 2001 and January 2019, patients that met at least one of the three abovementioned criteria (diagnostic codes of mycosis or aspergillosis, positive culture results of fungi, or the presence of fungi in the pathology report) for fungal rhinosinusitis diagnosis in our database were enrolled. After excluding patients aged <18 years, repeated data, and patients who were not admitted due to sinusitis, we included 1624 patients in our study, where 61% were female patients and 39% were male patients. The median age was 57.7 years (IQR, 25–75% = 48.9–66.3). The diagnosis of IFS was established in 59 patients, and the other 1565 patients were classified as NIFS. Table 1 shows the patient characteristics and laboratory results of our study. The IFS group was older and predominantly male compared with the NIFS group. Underlying comorbidities including diabetes mellitus and hematologic malignancies (lymphoma or leukemia) were more commonly seen in the IFS group, with significant differences. The lab data, such as white blood cell (WBC) count, hemoglobin (Hb), blood glucose, glycohemoglobin (HbA1c), C-reactive protein (CRP), albumin, and erythrocyte sedimentation rate (ESR), varied between the IFS and NIFS groups. In addition, the IFS group had a higher prevalence of fever upon triage, leukopenia, and severe neutropenia (absolute neutrophil count (ANC) level < 500). The NIFS group had a higher percentage of receiving sinus surgery (96.2%), less chances of repeated surgery, and a shorter time to surgery period compared with the IFS group. Lastly, the IFS group had more orbital complications, longer hospitalization, and a higher mortality rate.

#### 3.1.1. Non-Invasive Fungal Rhinosinusitis with or without Orbital Complications

IFS and NIFS patients were then subgrouped according to the presence of orbital complications. In the NIFS group, 18 out of 1565 patients (1.2%) presented orbital complications. Patients with or without orbital complications both showed female predominance (Table 2). At the time of diagnosis, the NIFS with orbital complications group was older. The lab data, including a higher WBC count, higher neutrophil count, higher blood glucose, and higher CRP level, were associated with orbital complications. A higher percentage (96.5%) of the patients with NIFS without orbital complications received sinus surgery. The time to surgery and length of hospital stay were significantly longer in the NIFS with orbital complications group. The NIFS without orbital complications group had three patients who died within six months, but their deaths were found not to be related to the FRS after reviewing the medical records.

#### 3.1.2. Invasive Fungal Rhinosinusitis with or without Orbital Complications

Of the IFS patients, 26 out of 59 presented orbital complications (44.1%). There was no difference in age, gender, smoking history, or other underlying diseases between patients with or without orbital complications (Table 3). The IFS with orbital complications group had a higher WBC count and ANC level compared to the IFS without orbital complications group. Around 90 percent of patients in both groups received sinus surgery. A longer interval between admission and surgery was observed in IFS patients without orbital complications. A higher rate of repeated surgery was found in IFS patients with orbital complications. In this multi-institutional study, the short-term mortality rate was 11.9 percent in IFS patients. Three and four patients died within six months in the IFS with orbital complications group and IFS without orbital complications group, respectively, without significant differences.

### 3.2. Fungal Types and Treatment in IFS

The fungal types were confirmed by the histopathology reports or the culture results in those who did not undergo surgery or a biopsy but had a high clinical suspicion of IFS. A total of 31 patients (52.5%) were identified as having aspergillosis, and 15 patients (25.4%) were affected by mucormycosis (Table 4). Aspergillosis was predominantly found in IFS patients without orbital complications, and mucormycosis was primarily found in IFS patients with orbital complications. The time to prescription of antifungal medication was longer in the IFS without orbital complications group. Around 90% of patients in both subgroups received surgical intervention. Further, 72.7% and 61.5% of patients received a sinusectomy or debridement in the IFS without orbital complications group and IFS with orbital complications group, respectively. Full-house surgery (30.8%) was performed relatively more frequently in the IFS with orbital complications group.

### 3.3. Clinical Symptoms and Radiological Findings in IFS

Regarding the sinus involvement in the radiological findings, IFS patients with orbital invasion showed a significantly higher prevalence of involvement in the posterior ethmoid sinus (84.6%) and sphenoid sinus (73.1%) (Table 5). IFS patients who presented facial pain, blurred vision, periorbital swelling, proptosis, ptosis, diplopia, and EOM limitation were at higher risk for orbital complications. Headache, the most common symptom among IFS patients, showed a trend of association with orbital complications but without statistical significance.

### 3.4. Survival Outcomes in IFS

Table 6 shows the results of the Cox regression models for the three-month and six-month overall survivals. A history of organ or hematopoietic cell transplantation had a significant prognostic effect on the survival outcomes (HR = 10.716, 95% CI: 1.926–59.633). Hematologic malignancy, including lymphoma and leukemia, showed a trend of poor prognosis but without statistically significant results. Lab data, including the WBC count, ANC level < 500, CRP level, and ESR, showed no statistically significant difference between the two subgroups. In contrast, surgical intervention and high hemoglobin (Hb) and albumin levels were recognized as positive predictors.

### 3.5. Orbital Complications in IFS

Orbital complications were presented in 26 patients (44.1%) with IFS (Table 5). Staged by Chandler’s classification, the most common orbital complication was cavernous sinus thrombosis (46.2%), followed by orbital cellulitis (30.8%). We further investigated the clinical factors for the prediction of orbital complications. After analysis through univariable logistic regression, patients’ demographics, lab data, and underlying diseases showed no statistical significance for determining orbital complications in IFS patients (Table 7).

In the radiological findings, involvement of the posterior ethmoid sinus (OR = 4.583, 95% CI: 1.291–16.266) and sphenoid sinus (OR = 3.684, 95% CI = 1.216–11.155) was found to be a risk factor of orbital complications. In addition, clinical presentations including facial pain (OR = 4.267, 95% CI = 1.420–12.824), periorbital swelling (OR = 51.174, 95% CI = 6.015–435.387), and blurred vision (OR = 23.466, 95% CI = 2.769–198.845) were also proposed as predictors for orbital complications.

## 4. Discussion

In the present study, we revealed that IFS patients had a significantly higher prevalence of diabetes mellitus and hematologic malignancy compared with NIFS patients. In addition, fever upon triage, leukopenia, WBC count, hemoglobin level, albumin level, blood glucose, HbA1c, CRP, and ESR also showed significant differences between IFS and NIFS patients. Lastly, significantly higher orbital complication rates, a prolonged length of hospitalization, and higher mortality rates were found in the IFS group. In a further analysis among NIFS patients, old age and lab data, including the WBC count, ANC, blood glucose, and CRP level, were associated with orbital complications. Moreover, NIFS patients with orbital complications had a lower percentage of receiving surgery, a longer time to surgery period, and a longer hospital stay. Among IFS patients, we found that a history of organ or bone marrow transplantation was a predictor of poor survival, while surgical intervention, a high Hb level, and a high albumin level were, conversely, related to improved survival. For the development of orbital complications, posterior ethmoid sinus involvement, sphenoid sinus involvement, facial pain, blurred vision, and periorbital swelling were presented as clinical predictors in IFS patients. Although several studies have investigated the risk factors of poor prognosis in IFS patients, only some case series and single-institution studies have discussed the risk factors of orbital complications in IFS patients [11,27,28].

Several risk factors have been proposed for the early differentiation of IFS from NIFS. Yin et al. conducted a study that included 283 patients, comparing IFS and NIFS patients with risk factors recognized as fever, unilateral facial swelling, pain, erythema, and orbital involvement [29]. Other clinical factors such as thrombocytopenia (platelet count < 8.1 × 10^5^ cells/mm^3^), neutropenia (low ANC level), a recent history of chemotherapy, diabetes mellitus, hematologic malignancy, and solid organ transplantation were also found to be associated with a higher risk of IFS [12,30,31,32,33]. The comparison between the IFS and NIFS groups in our study shows high consistency with the abovementioned finding. Most patients with IFS were admitted during the acute stage of the disease; thus, the IFS group had higher levels of inflammatory markers, including the WBC count, CRP, and ESR. Conversely, a higher percentage of IFS patients had comorbidities, such as diabetes mellitus and hematologic malignancy, which may have led to differences in blood glucose, HbA1c, Hb levels, albumin levels, and leukopenia among these two groups. Most of the NIFS patients visited the outpatient clinic due to chronic symptoms and were admitted for scheduled sinus surgery, which may have led to a higher percentage (96.2%) of receiving surgery. The IFS patients had higher chances of repeated surgery and a longer time to surgery period, which may be related to the difficulty in the diagnosis of IFS, conservative treatment during the early period of the hospital stay, and the possibility of repeated debridement. A slight male predominance (52.5%) was found in the IFS group, consistent with a previous systematic review by Turner et al. (57.3%) and the multi-institutional study by Wandell et al. (63%) [12,20]. On the other hand, the female predominance of 61.5% in the NIFS group is consistent with the results in several studies. Factors including indolent characteristics, a high prevalence of NIFS in older patients, and the long lifespan of the female gender may contribute to the female predominance in NIFS [34,35,36].

In NIFS patients, orbital complications were found to be associated with old age, a high WBC count, high blood glucose, and a high CRP level, which suggests poor general performance, immunosuppressive status, and severe systemic inflammation. NIFS patients with orbital complications had a lower percentage of receiving surgery and a longer time to surgery period compared with NIFS patients without orbital complications. NIFS patients with orbital complications often visited the emergency department during the acute stage of the disease, while other NIFS patients were hospitalized for scheduled sinus surgery. In addition, patients with orbital complications were often treated with conservative treatment first before surgery, which may have led to a longer time to surgery period. In IFS patients with orbital complications, the median time to surgery period was 1.5 days, significantly shorter than the median of 3 days in IFS patients without orbital complications. IFS patients with orbital complications frequently presented more severe symptoms upon the visit. These patients may receive surgery due to failure of the conservative treatment within 24 to 48 h. Conversely, IFS patients without orbital complications may need further examinations, such as culture reports, pathology results, radiological findings, or endoscopic findings, to proceed to surgery from conservative treatment. Patients with orbital complications in both the IFS and NIFS groups had a longer hospital stay. Mortality showed no significant difference between patients with or without orbital complications in both groups. 

In previous studies evaluating the relationship between fungal type and mortality, inconsistency was found across the studies. Most studies with a small cohort reported insignificant findings for mucormycosis as a risk factor [17,21,22,23]. In the systematic review conducted by Turner et al., which recruited 807 patients from 52 articles, the fungal type had no impact on overall survival [12]. Contrarily, Burton et al. conducted a nationwide retrospective study, which included 979 IFS patients and revealed mucormycosis as a factor associated with inpatient mortality (OR = 2.95, 95% CI = 2.00–4.34) [18]. In our study, fungal infection by mucormycosis showed no predictive effect on overall survival. The relatively lower mortality rate in our cohort and the high percentage of surgical intervention (100%) in mucormycosis may have contributed to this result. For the prediction of orbital invasion, mucormycosis showed a trend of higher risk compared to other fungal infections, but without statistical significance (OR = 2.382, 95% CI: 0.719–7.894, *p* = 0.156).

Hematologic malignancies are one of the most common underlying conditions in IFS patients. Previous studies have investigated hematologic disorders as a risk factor for poor survival outcomes [20,37], but inconsistent findings were still found in several studies with a small case number [17,22,23,28]. Burton et al. reported hematologic disorder as a factor associated with inpatient mortality (OR = 1.92, 95% CI: 1.08–3.39), while Turner et al. demonstrated that hematologic malignancy had no association with mortality in their meta-analysis [12,18]. Our study presented hematologic malignancy with a trend of increased risk for poor overall survival but without statistical significance (HR = 3.988, 95% CI: 0.727–21.881). As another condition that may also be considered for immunosuppression status, the history of organ or bone marrow transplantation was proposed as a risk factor for poor prognosis in our analysis. In previous studies, bone marrow transplantation as a predictor of poor survival was only reported in the multi-institutional study by Wandell et al. (HR = 2.5, 95% CI: 1.1–5.3) [20]. On the other hand, some studies investigated the role of solid organ transplantation as a survival predictor, but the result showed no statistical significance [12,22]. Although we revealed organ or hematopoietic cell transplantation as a prognostic factor for survival, only a small population of patients had a history of transplantation in our study. Further studies that recruit more hematopoietic transplantation patients may be needed to confirm the predictive role of hematopoietic cell transplantation in IFS patients. 

Sinus involvement in imaging studies has long been recognized as a factor for developing orbital complications in rhinosinusitis. Ethmoid sinus involvement was the most common finding related to orbital complications in previous studies [38,39,40]. Snidvongs et al. reported ethmoid sinus involvement as an independent factor for orbital complications in a case–control study that included 43 patients with acute rhinosinusitis (OR = 31.1, 95% CI: 2.3–430.6) [41]. Among patients with IFS, a higher prevalence of ethmoid sinus and sphenoid sinus origin was found in patients with orbital involvement [40]. A retrospective study by Twu et al. that included 38 patients further proposed IFS originating from the sphenoid sinus (OR = 21.875, 95% CI: 3.295–145.237) and posterior ethmoid sinus (OR = 14.0, 95% CI: 1.234–158.844) to have a higher risk of developing orbital complications [11]. Familiar theories of orbital invasion are the direct spread through the lamina papyracea to the orbit or the indirect pathway through the emboli along the ethmoid vein. Additionally, cavernous sinus thrombosis may result from the spreading of fungal infection through the posterolateral wall of the sphenoid sinus. In our study, we reported the involvement of the posterior ethmoid sinus and sphenoid sinus as an independent risk factor for developing orbital complications, in accordance with the findings of the previous literature. 

To achieve an accurate and early diagnosis under limited information upon the patient visit, the roles of initial presentations and patient characteristics are crucial for predicting orbital complications in IFS. In a previous study conducted by Twu et al., headache (OR: 16.8, 95% CI = 4.758–1627.70) and fever (RR = 8.25) were proposed as risk factors for orbital complications. In the present study, headache (73.1%), and fever (34.6%) were found with a higher prevalence in IFS patients with orbital complications but without significant differences. Instead, IFS patients with orbital complications in our study more commonly presented with blurred vision; periorbital swelling; facial pain; and other ophthalmologic symptoms (proptosis, ptosis, diplopia, and EOM limitation), with comparable prevalence to a previous meta-analysis and multi-institutional study [12,20]. Among these symptoms, periorbital swelling, facial pain, and blurred vision were demonstrated as risk factors for orbital complications in our study, which has not been mentioned in previous studies.

Laboratory data are also a valuable tool in research for predicting prognosis and orbital complications. Severe neutropenia with ANC < 500 was mentioned as a risk factor for poor survival in several studies [20,22,23]. Our study also demonstrated a similar trend in severe neutropenia (OR = 6.914, 95% CI: 0.769–62.165) but without statistical significance, which may be related to the small sample size of patients recorded with severe neutropenia in the present cohort. In our analysis of predictors of overall survival, high Hb (OR = 0.725, 95% CI: 0.541–0.972) and albumin levels (OR = 0.117, 95% CI: 0.017–0.785) were determined as positive predictors of overall survival, which has not been mentioned in previous analyses. These two factors are often assumed to be the index for general performance status, and a lower level of these two markers may predispose patients to the frailness of fulminant and systemic infection. Further analysis conducted on the cut-off values of these predictive factors may provide a better prognostic value. Surgical intervention as a positive prognostic factor has been mentioned in several studies, and our study confirmed the conclusions in the previous literature [12,20,42].

In previous studies conducted on IFS patients, the long-term mortality rate ranged from 18% to 64.9% [12,16,17,18,19]. We demonstrated 11.9% of mortality within six months in our study, which is comparable to the result of inpatient mortality of around 15.8% in the nationwide study conducted by Burton et al. [18]. Short-term mortality may reflect precise disease-related deaths and may also be the reason for the lower mortality rate in our study. The prevalence of orbital complications was 44.1% in our study, consistent with a previous systematic review by Turner et al. [12].

To our knowledge, this is the largest cohort used to compare the characteristics of IFS and NIFS, as well as the orbital complications and survival outcomes in different subgroups. Although this is a multi-institutional study consisting of tertiary centers and regional hospitals, it has several limitations. Owing to the rarity of IFS, the sample size was small in the different subgroups. The retrospective nature of this study and incomplete data were also notable limitations. Although we tried to minimize the bias by using strict inclusion criteria and repeatedly reviewing the medical records, bias may still have occurred during the process of recruitment from the database. Further prospective, nationwide studies are needed to evaluate predictors of mortality and orbital complications in IFS patients.

## 5. Conclusions

This is the first multi-institutional study discussing factors predicting orbital complications in IFS and NIFS patients. In NIFS patients, orbital complications were found to be associated with old age, a high WBC count, high blood glucose, and a high CRP level. For the risk factors of orbital complications in IFS patients, posterior ethmoid sinus involvement, sphenoid sinus involvement, facial pain, blurred vision, and periorbital swelling were recognized as predictors. Among IFS patients, a history of organ or hematopoietic cell transplantation was a risk factor for poor survival, while surgery and high Hb and albumin levels were, conversely, related to improved survival. Close observation of IFS patients using the abovementioned factors upon the first visit may contribute to better orbital and survival outcomes.

## Figures and Tables

**Table 1 jof-08-01239-t001:** Demographic data, laboratory results, surgical outcomes, morbidity, and mortality between the IFS and NIFS groups.

	All Patients (*n* = 1624)	IFS (*n* = 59)	NIFS (*n* = 1565)	*p*-Value
**Age** (median (IQR))	57.7 (48.9–66.3)	63.7 (55.2–72.6)	57.5 (48.8–66.0)	**0.002**
**Gender** (No (%))				**0.030**
Male	633 (39.0%)	31 (52.5%)	602 (38.5%)	
Female	991 (61.0%)	28 (47.5%)	963 (61.5%)	
**Underlying diseases** (No (%))				
Asthma	119 (7.3%)	4 (6.8%)	115 (7.3%)	1.000
COPD	158 (9.7%)	7 (11.9%)	151 (9.6%)	0.573
ESRD	26 (1.6%)	3 (5.1%)	23 (1.5%)	0.065
Liver cirrhosis	30 (1.8%)	1 (1.7%)	29 (1.9%)	1.000
Diabetes mellitus	246 (15.1%)	18 (30.5%)	228 (14.6%)	**0.001**
HIV/AIDS	18 (1.1%)	1 (1.7%)	17 (1.1%)	0.488
Transplantation status *	26 (1.6%)	3 (5.1%)	23 (1.5%)	0.065
Hematologic malignancy	20 (1.2%)	7 (11.9%)	13 (0.8%)	**<0.001**
**Lab data** (median (IQR))				
Fever (No (%))	18 (1.1%)	18 (30.5%)	0 (0.0%)	**<0.001**
WBC (10^3^/µL)	6.5 (5.5–8.0)	9.6 (6.6–15.1)	6.5 (5.5–7.9)	**<0.001**
Hemoglobin (g/L)	134 (123–144)	117 (106–135)	134 (124–145)	**<0.001**
Platelets (10^3^/µL)	236.0 (198.0–281.0)	224.0 (147.0–303.0)	236.0 (199.0–281.0)	0.321
Leukopenia (No (%)) **	9 (0.6%)	2 (3.4%)	7 (0.4%)	**0.027**
ANC (/µL)	3850.2 (3079.3–4992.4)	7488.8 (3978.5–11379.2)	3822.6 (3056.4–4858.7)	**<0.001**
Severe neutropenia ***	2 (0.1%)	2 (3.4%)	0 (0.0%)	**0.002**
Albumin (g/dL)	4.4 (4.1–4.6)	3.6 (3.2–4.1)	4.4 (4.1–4.6)	**<0.001**
Glucose (mg/dL)	114.0 (98.0–145.0)	163.5 (127.5–301.0)	113.0 (97.0–143.0)	**<0.001**
HbA1c (%)	6.2 (5.6–7.4)	7.2 (6.2–10.4)	6.1 (5.6–7.2)	**0.001**
CRP (mg/L)	7.3 (1.9–38.5)	83.2 (5.7–180.7)	6.5 (1.9–27.7)	**<0.001**
ESR (mm/h)	13.0 (8.0–29.0)	36.0 (13.0–58.0)	13.0 (8.0–25.0)	**0.003**
**Surgical outcomes**				
Surgery performed (No (%))	1559 (96.0%)	53 (89.8%)	1506 (96.2%)	**0.028**
Number of procedures (times, median (IQR))	1 (1–1)	1 (1–2)	1 (1–1)	**<0.001**
Time to surgery (days, median (IQR))	1 (1–1)	2 (1–5)	1 (1–1)	**<0.001**
**Length of hospital stay** (days, median (IQR))	4 (3–5)	26 (9–42)	4 (3–5)	**<0.001**
**3 months mortality** (No (%))	7 (0.4%)	5 (8.5%)	2 (0.1%)	**<0.001**
**6 months mortality** (No (%))	10 (0.6%)	7 (11.9%)	3 (0.2%)	**<0.001**

Abbreviations: IQR, interquartile range; IFS, invasive fungal rhinosinusitis; NIFS, non-invasive fungal rhinosinusitis; COPD, chronic obstructive pulmonary disease; CKD, chronic kidney disease; ESRD, end-stage renal disease; AIDS, acquired immunodeficiency syndrome; ANC, absolute neutrophil count; WBC, white blood cell; CRP, C-reactive protein; ESR, erythrocyte sedimentation rate; HbA1c, glycohemoglobin. Bold text indicates statistical significance (*p* < 0.05). * Transplantation status: including organ transplantation and hematopoietic cell transplantation. ** Leukopenia: WBC count < 3000. *** Severe neutropenia: ANC < 500.

**Table 2 jof-08-01239-t002:** Demographic data, laboratory results, surgical outcomes, morbidity, and mortality between the NIFSsOC and NIFSwOC groups.

	NIFSsOC (*n* = 1547)	NIFSwOC (*n* = 18)	*p*-Value
**Age** (median (IQR))	57.4 (48.7–66.0)	64.7 (60.0–69.3)	**0.009**
**Gender** (No (%))			0.970
Male	595 (38.5%)	7 (38.9%)	
Female	952 (61.5%)	11 (61.1%)	
**Underlying diseases** (No (%))			
Diabetes mellitus	225 (14.5%)	3 (16.7%)	0.738
HIV/AIDS	16 (1.0%)	1 (5.6%)	0.179
Transplantation status *	23 (1.5%)	0 (0.0%)	1.000
Hematologic malignancy	13 (0.8%)	0 (0.0%)	1.000
**Lab data** (median (IQR))			
WBC (10^3^/µL)	6.5 (5.5–7.8)	9.9 (8.3–13.2)	**<0.001**
Hemoglobin (g/L)	134 (124–145)	132 (116–141)	0.408
Platelets (10^3^/µL)	235.0 (199.0–280.0)	282.5 (192.0–312.0)	0.118
ANC (/µL)	3794.4 (3048.5–4800.0)	8238.2 (6242.6–10861.8)	**<0.001**
Albumin (g/dL)	4.4 (4.1–4.6)	3.9 (3.2–4.6)	0.374
Glucose(mg/dL)	112.6 (97.0–142.5)	164.5 (108.0–271.0)	**0.023**
HbA1c (%)	6.1 (5.6–7.2)	6.6 (5.6–9.0)	0.289
CRP (mg/L)	6.2 (1.8–25.7)	57.3 (7.2–250.4)	**0.009**
ESR (mm/h)	13.0 (8.0–25.0)	11.0 (5.0–23.0)	0.590
**Surgical outcomes**			
Surgery performed (No (%))	1493 (96.5%)	13 (72.2%)	**<0.001**
Number of procedures (times, median (IQR))	1 (1–1)	1 (1–1)	0.709
Time to surgery (days, median (IQR))	1 (1–1)	2 (0–4)	**0.011**
**Length of hospital stay** (days, median (IQR))	3 (3–4)	8.5 (6–19)	**<0.001**
**3 months mortality** (No (%))	2 (0.1%)	0 (0.0%)	1.000
**6 months mortality** (No (%))	3 (0.2%)	0 (0.0%)	1.000

Abbreviations: IQR, interquartile range; NIFSsOC, NIFS without orbital complications; NIFSwOC, NIFS with orbital complications; AIDS, acquired immunodeficiency syndrome; ANC, absolute neutrophil count; WBC, white blood cell; CRP, C-reactive protein; ESR, erythrocyte sedimentation rate; HbA1c, glycohemoglobin. Bold text indicates statistical significance (*p* < 0.05). * Transplantation status: including organ transplantation and hematopoietic cell transplantation.

**Table 3 jof-08-01239-t003:** Demographic data, laboratory results, surgical outcomes, morbidity, and mortality between the IFSsOC and IFSwOC groups.

	IFSsOC (*n* = 33)	IFSwOC (*n* = 26)	*p*-Value
**Age** (median (IQR))	63.7 (55.2–73.8)	65.9 (55.9–71.0)	0.982
**Gender** (No (%))			0.219
Male	15 (45.5%)	16 (61.5%)	
Female	18 (54.5%)	10 (38.5%)	
**Underlying diseases**			
Diabetes mellitus	10 (30.3%)	8 (30.8%)	0.969
HIV/AIDS	1 (3.0%)	0 (0.0%)	1.000
Transplantation status *	2 (6.1%)	1 (3.8%)	1.000
Hematologic malignancy	6 (18.2%)	1 (3.8%)	0.121
**Lab data** (days, median (IQR))			
Fever (No (%))	9 (27.3%)	9 (34.6%)	0.543
WBC (10^3^/µL)	8.6 (6.0–12.5)	11.4 (8.7–15.2)	**0.046**
Hemoglobin (g/L)	115 (102–130)	121 (109–137)	0.235
Platelets (10^3^/µL)	212.0 (158.0–280.0)	236.0 (143.0–323.0)	0.410
ANC (/µL)	5959.0 (3611.4–7815.3)	10394.4 (6947.2–12819.9)	**0.011**
Albumin (g/dL)	3.6 (2.7–3.9)	3.5 (3.5–4.1)	0.752
Glucose (mg/dL)	151.5 (124.0–294.0)	214.5 (139.0–368.0)	0.122
HbA1c (%)	7.1 (6.2–10.4)	7.2 (7.2–7.9)	0.487
CRP (mg/L)	20.6 (2.2–101.6)	123.6 (17.9–256.0)	0.062
ESR (mm/h)	44.0 (11.0–59.0)	30.0 (16.0–53.0)	0.970
**Surgical outcomes**			
Surgery performed (No (%))	29 (87.9%)	24 (92.3%)	0.685
Number of procedures (times, median (IQR))	1.0 (1.0–1.0)	1.0 (1.0–2.0)	0.089
Time to surgery (days, median (IQR))	3.0 (1.0–7.0)	1.5 (0.0–3.0)	**0.024**
**Length of hospital stay** (days, median (IQR))	16 (5–38)	36.5 (24–49)	**0.008**
**3 months mortality** (No (%))	4 (12.1%)	1 (3.8%)	0.372
**6 months mortality** (No (%))	4 (12.1%)	3 (11.5%)	1.000

Abbreviations: IQR, interquartile range; IFSsOC, IFS without orbital complications; IFSwOC, IFS with orbital complications; AIDS, acquired immunodeficiency syndrome; ANC, absolute neutrophil count; WBC, white blood cell; CRP, C-reactive protein; ESR, erythrocyte sedimentation rate; HbA1c, glycohemoglobin. Bold text indicates a statistical significance (*p* < 0.05). * Transplantation status: including organ transplantation and hematopoietic cell transplantation.

**Table 4 jof-08-01239-t004:** Comparison of the fungal type and surgery between the IFSsOC group and IFSwOC group.

	IFSsOC (*n* = 33)	IFSwOC (*n* = 26)	*p*-Value
**Time to antifungal medication** (days, median (IQR))	12 (4–22)	3.5 (2–9)	**0.012**
**Fungal type** (No (%))			0.355
Mucormycosis	6 (18.2%)	9 (34.6%)	
Aspergillosis	19 (57.6%)	12 (46.2%)	
Other	8 (24.2%)	5 (19.2%)	
**Surgery** (No (%))			0.366
Sinusectomy	24 (72.7%)	16 (61.5%)	
Full-house surgery	5 (15.2%)	8 (30.8%)	
No surgery	4 (12.1%)	2 (7.7%)	

Abbreviations: IQR, interquartile range; IFSsOC, invasive fungal rhinosinusitis without orbital complications; IFSwOC, invasive fungal rhinosinusitis with orbital complications. Bold text indicates a statistical significance (*p* < 0.05).

**Table 5 jof-08-01239-t005:** Grading of orbital complications, radiological findings, and symptoms between the IFSsOC group and IFSwOC group.

	IFSsOC (*n* = 33)	IFSwOC (*n* = 26)	*p*-Value
**Orbital complications** (No (%))			
Stage 1		2 (7.7%)	
Stage 2		8 (30.8%)	
Stage 3		1 (3.8%)	
Stage 4		3 (11.5%)	
Stage 5		12 (46.2%)	
**Sinus involvement in CT** (No (%))			
Frontal sinus	5 (15.2%)	4 (15.4%)	1.000
Maxillary sinus	25 (75.8%)	20 (76.9%)	0.917
Anterior ethmoid sinus	20 (60.6%)	20 (76.9%)	0.183
Posterior ethmoid sinus	18 (54.5%)	22 (84.6%)	**0.014**
Sphenoid sinus	14 (42.4%)	19 (73.1%)	**0.019**
**Symptoms** (No (%))			
Headache	17 (51.5%)	19 (73.1%)	0.092
Facial pain	9 (27.3%)	16 (61.5%)	**0.008**
Facial swelling	5 (15.2%)	8 (30.8%)	0.151
Fever	9 (27.3%)	9 (34.6%)	0.543
Blurred vision	1 (3.0%)	11 (42.3%)	**<0.001**
Periorbital swelling	1 (3.0%)	16 (61.5%)	**<0.001**
Proptosis	0 (0.0%)	5 (19.2%)	**0.013**
Loss of cheek sensation	5 (15.2%)	3 (11.5%)	1.000
Ptosis	0 (0.0%)	7 (26.9%)	**0.002**
Diplopia	0 (0.0%)	9 (34.6%)	**<0.001**
EOM limitation	0 (0.0%)	10 (38.5%)	**<0.001**
Conscious disturbance	2 (6.1%)	6 (23.1%)	0.122

Abbreviations: IQR, interquartile range; IFSsOC, invasive fungal rhinosinusitis without orbital complications; IFSwOC, invasive fungal rhinosinusitis with orbital complications; EOM, extraocular muscle. Bold text indicates a statistical significance (*p* < 0.05).

**Table 6 jof-08-01239-t006:** Univariate Cox regression analysis of prognostic factors of survival in IFS patients.

	3-Month Hazard Ratio (95% CI)	*p*-Value	6-Month Hazard Ratio (95% CI)	*p*-Value
**Age**	0.998 (0.944, 1.056)	0.953	0.991 (0.942, 1.041)	0.709
**Gender**				
Male	1.355 (0.226, 8.108)	0.739	1.825 (0.334, 9.967)	0.487
**Underlying disease**				
Diabetes mellitus	1.519 (0.254, 9.088)	0.647	1.149 (0.210, 6.272)	0.873
Transplantation status *	12.444 (2.079, 74.475)	**0.006**	10.716 (1.926, 59.633)	**0.007**
Hematologic malignancy	4.952 (0.828, 29.638)	0.080	3.988 (0.727, 21.881)	0.111
**Sinus surgery performed**	0.075 (0.013, 0.452)	**0.005**	0.103 (0.021, 0.514)	**0.006**
**Any orbital complication**	0.317 (0.035, 2.839)	0.305	0.630 (0.115, 3.441)	0.594
**Lab data**				
WBC (10^3^/µL)	0.936 (0.767, 1.141)	0.510	0.942 (0.791, 1.121)	0.499
Hemoglobin (g/L)	0.668 (0.488, 0.915)	**0.012**	0.725 (0.541, 0.972)	**0.031**
Leukopenia ***	1.106 (0.873, 1.401)	0.402	1.079 (0.856, 1.360)	0.520
ANC (/µL)	1.000 (1.000, 1.000)	0.745	1.000 (1.000, 1.000)	0.697
Severe neutropenia **	8.500 (0.884, 81.715)	0.064	6.914 (0.769, 62.165)	0.084
Albumin (g/dL)	0.212 (0.030, 1.489)	0.119	0.117 (0.017, 0.785)	**0.027**
Glucose (mg/dL)	0.998 (0.992, 1.005)	0.601	0.997 (0.989, 1.005)	0.414
HbA1c (%)	0.912 (0.571, 1.457)	0.700	0.912 (0.571, 1.457)	0.700
CRP (mg/L)	1.003 (0.993, 1.012)	0.612	1.005 (0.996, 1.013)	0.269
ESR (mm/h)	1.022 (0.973, 1.074)	0.389	1.022 (0.973, 1.074)	0.389
**Fungal type**				
Mucormycosis	-	-	0.852 (0.053, 13.626)	0.910
Aspergillosis	1.677 (0.187, 15.008)	0.644	1.680 (0.188, 15.030)	0.643
**Sinus involvement in CT**				
Maxillary sinus	1.244 (0.139, 11.133)	0.845	1.581 (0.185, 13.535)	0.676
Sphenoid sinus	3.151 (0.352, 28.195)	0.305	4.079 (0.476, 34.945)	0.200
**Symptoms**				
Headache	2.556 (0.286, 22.864)	0.401	3.291 (0.384, 28.190)	0.277
Facial pain	0.340 (0.038, 3.042)	0.335	0.267 (0.031, 2.285)	0.228
Facial swelling	2.359 (0.394, 14.118)	0.347	1.812 (0.332, 9.900)	0.493
Fever	3.417 (0.571, 20.447)	0.178	4.672 (0.855, 25.523)	0.075
Periorbital swelling	0.618 (0.069, 5.526)	0.666	0.492 (0.057, 4.209)	0.517
Loss of cheek sensation	1.594 (0.178, 14.259)	0.677	1.272 (0.149, 10.888)	0.826

Abbreviations: IQR, interquartile range; IFS, invasive fungal rhinosinusitis; ANC, absolute neutrophil count; WBC, white blood cell; CRP, C-reactive protein; ESR, erythrocyte sedimentation rate; HbA1c, glycohemoglobin. Bold text indicates a statistical significance (*p* < 0.05). * Transplantation status: including organ transplantation and hematopoietic cell transplantation. ** Severe neutropenia: ANC < 500. *** Leukopenia: WBC count < 3000.

**Table 7 jof-08-01239-t007:** Univariable logistic regression for factors predicting orbital complications in IFS.

	Odds Ratio	95% CI	*p*-Value
**Age**	0.999	(0.966, 1.033)	0.962
**Gender**			
Male	1.920	(0.675, 5.464)	0.222
Female	Reference		
**Underlying disease**			
Diabetes mellitus	1.022	(0.335, 3.120)	0.969
Transplantation status *	0.620	(0.053, 7.240)	0.703
Hematologic malignancy	0.180	(0.020, 1.602)	0.124
**Lab data**			
CRP > 10	2.045	(0.463, 9.033)	0.345
ESR > 45	0.600	(0.097, 3.720)	0.583
HbA1c	0.982	(0.701, 1.377)	0.918
**Fungal type**			
Mucormycosis	2.382	(0.719, 7.894)	0.156
Other	Reference		
**Sinus involvement in CT**			
Frontal sinus	1.018	(0.244, 4.248)	0.980
Maxillary sinus	1.067	(0.318, 3.580)	0.917
Anterior ethmoid sinus	2.166	(0.687, 6.834)	0.187
Posterior ethmoid sinus	4.583	(1.291, 16.266)	**0.018**
Sphenoid sinus	3.684	(1.216, 11.155)	**0.021**
**Symptoms**			
Headache	2.554	(0.847, 7.697)	0.096
Facial pain	4.267	(1.420, 12.824)	**0.010**
Facial swelling	2.489	(0.703, 8.814)	0.158
Fever	1.412	(0.464, 4.298)	0.544
Blurred vision	23.466	(2.769, 198.845)	**0.004**
Periorbital swelling	51.174	(6.015, 435.387)	**<0.001**
Loss of cheek sensation	0.730	(0.158, 3.387)	0.688
Conscious disturbance	4.650	(0.853, 25.356)	0.076

Abbreviations: IQR, interquartile range; IFS, invasive fungal rhinosinusitis; NIFS, non-invasive fungal rhinosinusitis; CRP, C-reactive protein; ESR, erythrocyte sedimentation rate; HbA1c, glycohemoglobin. * Transplantation status: including organ transplantation and hematopoietic cell transplantation.

## Data Availability

Data are available upon reasonable request.

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
