# Peer review of "A Multi-Institutional Database Review of Orbital Complications and Survival Outcomes in Adult Patients with Invasive or Non-Invasive Fungal Rhinosinusitis"

_jof, 2022, doi:10.3390/jof8121239_

Round 1
Reviewer 1 Report
Chiang and colleagues submit a multi-institutional database review of orbital complications and survival outcomes of patients with fungal rhinosinusitis.
Comments:
- Since the study excluded persons under 18 years of age, please change the title to indicate that this is a study of adult patients.
- What can explain the male predominance of invasive infections?
- In table 1, the time to surgery is “1”. One what? One day? One week? What is the time from?
- In table 1, the length of hospital stay is “4”. Four what? Four days?
- Did the patients with NIFS have more surgeries because they lived long enough to get surgery? Does that indicate a bias in the way the data were analyzed?
- The term “bone marrow transplant” is out of date, as sometimes it could be umbilical cord blood or peripheral blood stem cells that are infused. Would replace each instance of “bone marrow transplant” with “hematopoietic cell transplant”.
- Introduction: Add the correct reference for Hora in 1965.
- Reference 6 had semicolons instead of commas between author listings, and the second author of M O’Brien is not correctly spelled out. Please update.
Author Response
Point 1: Since the study excluded persons under 18 years of age, please change the title to indicate that this is a study of adult patients.
Response 1: Thank you for the comment on a more precise title. We modified the title to “A Multi-institutional Database Review of Orbital Complications and Survival Outcomes in Adult patients with Invasive or Non-Invasive Fungal Rhinosinusitis”.
Point 2: What can explain the male predominance of invasive infections?
Response 2: The male predominance of invasive fungal sinusitis was also noted in the systemic review by Turner, et al. (57.3%) and the multi-institutional study by Wandell et al. (63%) but without further explanation in their studies. A slightly higher percentage of male patients (52.5%) was also found in the IFS group, consistent with previous studies. The difference in the number of male and female patients was not that obvious in our study (Female: Male = 31:28). The female predominance of around 64% in NIFS was consistently reported in several studies. [1, 2] Factors including indolent characteristics, high prevalence of NIFS in older patients, and the long life-spanning of the female gender may contribute to the condition of female predominance in NIFS. Both the slight male predominance in IFS and the female predominance in NIFS led to the statistical significance in Table 1. We added further discussion of these findings in lines 268-273
References
- Ferguson, B. J. (2000). Fungus balls of the paranasal sinuses. Otolaryngologic Clinics of North America, 33(2), 389-398.
- Klossek, J. M., Serrano, E., Péloquin, L., Percodani, J., Fontanel, J. P., & Pessey, J. J. (1997). Functional endoscopic sinus surgery and 109 mycetomas of paranasal sinuses. The laryngoscope, 107(1), 112-117.
Point 3: In table 1, the time to surgery is “1”. One what? One day? One week? What is the time from?
Response 3: The unit of the time to surgery is “day,“ and we corrected the omission in all tables.
Point 4: In table 1, the length of hospital stay is “4”. Four what? Four days?
Response 4: The unit of the length of hospital stay is “day, “ and we corrected the omission in all tables.
Point 5: Did the patients with NIFS have more surgeries because they lived long enough to get surgery? Does that indicate a bias in the way the data were analyzed?
Response 5: We had a mistake in the description of surgical outcomes between the NIFS group and the IFS group. Most of the NIFS group patients visited the outpatient clinic due to chronic symptoms and were admitted for scheduled sinus surgery, which may lead to a higher percentage (96.2%) of receiving surgery. The IFS groups patient had more chances of repeated surgery and a longer time to surgery period, according to the Table 1. The difficulty in diagnosis of IFS, conservative treatment during the early period of hospital stay, and the possibility of repeated debridement may lead to the results. We revised the descriptions in result section (lines 155-156 ) and added further discussion in lines 283-288
Point 6: The term “bone marrow transplant” is out of date, as sometimes it could be umbilical cord blood or peripheral blood stem cells that are infused. Would replace each instance of “bone marrow transplant” with “hematopoietic cell transplant”
Response 6: Thank you for the incisive comment. we replaced the “bone marrow transplant” with “hematopoietic cell transplant” according to your advice.
Point 7: Introduction: Add the correct reference for Hora in 1965.
Response 7: We added the study by Hora in 1965 in the reference.
Point 8: Reference 6 had semicolons instead of commas between author listings, and the second author of M O’Brien is not correctly spelled out. Please update.
Response 8: We corrected the mis-spelling of the second author and the mistake of using semicolons.
Reviewer 2 Report
In this manuscript the authors have performed a multi-institutional database review study of Invasive and Non-invasive fungal rhinosinusitits with and without orbital complications in Taiwan. The science is robust and the study is well performed. The conclusions are supported by the data and is presented and analysed appropriately. There are several minor comments that need addressing before publication;
- L34-35 "elevating prevalence" the references do not support this statement, a study in time needs to be cited to support such a statement or this needs rephrasing.
- L43 "by Hora in 1965" then cite this study.
- L52-53 "previous limited studies" if you are claiming multiple studies then cite multiple sources.
- L55-59 Training of clinicians should be mentioned as a factor as well.
- L85 "recruited" patients were not recruited, but "data from 1624 patients was obtained".
- 2.3 Statistical analysis. It would be useful to know which statistics are performed per dataset or comparison. Can this be included either in the methods or as a column in the tables?
- Throughout the results, the authors should comments on comparators where significance is found but sample sizes are small. This can bias the data.
- Table 4: How was species level confirmed? Just visually or by molecular methods. Mucor and Aspergillus look like a lot of other things. Moree clarity needed here
- Section 3.2 make sure to italicise and capitalise species names.
Author Response
Point 1: L34-35 "elevating prevalence" the references do not support this statement, a study in time needs to be cited to support such a statement or this needs rephrasing.
Response 1: Thank you for the comment on the imprecise reference. We revised the reference to studies supporting the rising prevalence of fungal rhinosinusitis.
Point 2: L43 "by Hora in 1965" then cite this study.
Response 2: We added the study by Hora in 1965 in the reference.
Point 3: L52-53 "previous limited studies" if you are claiming multiple studies then cite multiple sources.
Response 3: It was our mistake only to cite a single study while using the word "studies" in our manuscripts. We added other references discussed about the orbital involvement in radiological findings in NIFS.
Point 4: L55-59 Training of clinicians should be mentioned as a factor as well.
Response 4: Thank you for the comment. We added the experience of clinicians as a factor in the manuscript in L65-70.
Point 5: L85 "recruited" patients were not recruited, but "data from 1624 patients was obtained"
Response 5: We revised the sentence as recommended in L98-100.
Point 6: 2.3 Statistical analysis. It would be useful to know which statistics are performed per dataset or comparison. Can this be included either in the methods or as a column in the tables?"
Response 6: We described the statistics performed in each table in section 2.3 , and also mentioned it in the title of the Table 6 and Table 7.
Point 7: Throughout the results, the authors should comments on comparators where significance is found but sample sizes are small. This can bias the data.
Response 7: Firstly, the unit of several variables, including the number of procedures, time to surgery, and length of hospital stay, was median (IQR). It was our fault that the units were not marked clearly, which may lead to misunderstanding as a small sample size. We added the unit of these variables in Table 1 to Table 3. Due to ther rarity of IFS, the patient number of the IFS group and some variables were relatively small. We removed some variables with a patient number of less than ten ( All of them showed no statistical significance) to avoid possible bias in interpretation. Another variable with statistical significance but small sample size is “transplantation status”, we further discussed it and the possible bias in L338-346.
Point 8: Table 4: How was species level confirmed? Just visually or by molecular methods. Mucor and Aspergillus look like a lot of other things. More clarity needed here
Response 8: Most of the fungal species in our study were approved in histopathologic findings either by biopsy or surgery. Those with high clinical suspicion of IFS, including ophthalmologic complications or neurological deficits, but without biopsy or surgery had fungal species confirmed by culture reports. We revised the section 3.2. for better clarity.
Point 9: Section 3.2 make sure to italicise and capitalise species names.
Response 9: We italicised and capitalised the species names in section 3.2.